Estimating internal dissolved methane loading in rivers using a mass balance approach

http://orcid.org/0000-0002-6644-1701 Tsuchiya Kenji tsuchiya.kenji@nies.go.jp
Miura Shingo
Kohzu Ayato
Regional Environment Conservation Division, National Institute for Environmental Studies , Tsukuba, Ibaraki , Japan
Hornibrook Edward
Electronic publication date: 2025 Oct 15
Publication date: 2025
Volume: 13
Electronic Location ID: e20238
Received 2025 Apr 25; Accepted 2025 Sep 24
Copyright: © 2025 Tsuchiya et al.
Copyright year: 2025
Copyright holder: Tsuchiya et al.
License: This is an open access article distributed under the terms of the Creative Commons Attribution License, which permits unrestricted use, distribution, reproduction and adaptation in any medium and for any purpose provided that it is properly attributed. For attribution, the original author(s), title, publication source (PeerJ) and either DOI or URL of the article must be cited.
License URL: https://creativecommons.org/licenses/by/4.0/

Keywords: Riverbed, Irrigation weir, Methane oxidation, Emission to the atmosphere, Mass balance, Monitoring, Climate change

Funding: Japan Society for the Promotion of Science (JSPS) KAKENHI JP23K11412 Environmental Restoration and Conservation Agency JPMEERF20232002 The present study was supported by Japan Society for the Promotion of Science (JSPS) KAKENHI Grant Number JP23K11412 and Environmental Restoration and Conservation Agency Grant Number JPMEERF20232002. There was no additional external funding received for this study. The funders had no role in study design, data collection and analysis, decision to publish, or preparation of the manuscript.

==============================
The dynamics of dissolved methane concentration in rivers are influenced by losses through atmospheric emission and microbial oxidation, and by gains from internal (e.g., riverbed sediment) and external (e.g., groundwater) sources. Reducing riverine methane emissions, a key strategy for mitigating global warming, requires decreasing both internal and external loadings. To develop effective mitigation measures, it is essential to quantify these loadings separately. In this study, we estimated the internal methane loading in a river using a mass balance approach. We focused on river reaches without tributary inflow or significant discharge changes, assuming negligible external methane loading. Sampling was conducted at upstream and downstream sites of two short reaches (2.2 and 4.4 km) in the Kokai River, Japan, during 2022–23. Dissolved methane concentrations ranged from 237 ± 19 to 1,271 ± 6 nmol L–1, with changes from upstream to downstream ([CH4]downstream–[CH4]upstream) varying from –113 to 363 nmol L–1 . Methane oxidation rates and diffusive emission fluxes to the atmosphere were −1.2 ± 0.8~66 ± 19 µmol m–2 h–1 and 32 ± 10~199 ± 149 µmol m–2 h–1, respectively. The net flux of dissolved methane from the riverbed to the water varied from −33 to 160 µmol m–2 h–1. Compared to conventional methods, including benthic chambers and peeper sampling with model simulation, this approach is simple and facilitates methane flux measurements across multiple sites and diverse environmental gradients. By integrating the estimates from river reaches, the proposed method is applicable to large-scale assessments of internal methane loading in river systems.

Introduction

Aquatic ecosystems contribute significantly to global methane emissions, accounting for nearly half of the methane released into the atmosphere (Rosentreter et al., 2021). Among them, freshwater systems are considered major contributors; however, the role of rivers and streams in the global methane cycle remains inadequately quantified. Recent studies estimate methane flux from rivers at 27.9 Tg CH4 year–1 (Rocher-Ros et al., 2023), a value comparable to that of lakes and reservoirs (Johnson et al., 2021, 2022). Reducing methane emissions from rivers to the atmosphere has the potential to contribute to mitigating climate change.

Methane in rivers is supplied primarily via two pathways: internal production within the riverbed (i.e., sediment-water interface flux) and lateral input from external sources such as groundwater, wetlands, or soils (Fig. 1). These two pathways differ in the site and mechanism of methane generation, which in turn influence CH4 dynamics in rivers (Deirmendjian et al., 2019; Blackburn & Stanley, 2021; Rocher-Ros et al., 2023). For instance, although forest groundwater contains higher CH4 concentrations than agricultural groundwater, CH4 concentrations are often higher in agricultural streams due to enhanced sedimentary production driven by eutrophication (Deirmendjian et al., 2019). Thus, the development of pathway-specific strategies is essential for effectively mitigating methane emissions from rivers. Strategies for decreasing internal production and mitigating CH4 flux include minimizing organic matter sedimentation in riverbeds, enhancing oxygen supply, and improving water quality management. In contrast, mitigating external contributions involves wetland management and restoration, optimizing land use within watersheds, and establishing vegetation buffer zones.

Figure 1 Illustration of a simple mass balance model at the reach scale in a river.

Changes in CH4 concentration within a reach (from upstream to downstream) in the water column are influenced by internal loading from the riverbed (Finternal), external loading from sources outside the river (Fexternal), oxidation of CH4 in the water column (Foxidation), and outgassing at the river surface (Foutgassing). If the reach is sufficiently short such that the flow rates at the upstream and downstream ends are nearly identical, Fexternal can be assumed to be zero. This assumption allows Finternal to be estimated using a mass balance approach.

While external loading from surface waters can often be quantified through measurements of flow and concentration in tributaries, estimating groundwater-derived inputs is more labor-intensive and less feasible. In cases where river discharge remains relatively constant along a reach, external loading may be negligible. Under such conditions, it becomes possible to estimate internal methane loading using a mass balance approach based on concentration differences between upstream and downstream points (Fig. 1).

Mass balance methods have been widely used to estimate internal loading (diffusion of CH4 from sediment) in lentic systems (D’Ambrosio & Harrison, 2022), but their application in lotic systems remains limited. To date, only a few studies have attempted to quantify internal methane flux in rivers, using methods such as benthic chambers and mass balance approaches (Rovelli et al., 2022), modeling approaches combined with pore water sampling (Chen et al., 2023; Michaelis et al., 2024), and mass balance frameworks supported by stable isotope analysis (Balathandayuthabani et al., 2024). Although robust, these methods face several limitations: they often disturb natural flow conditions, require extended incubation times, and are difficult to deploy in deep or fast-flowing rivers. Moreover, high spatial heterogeneity in riverbed conditions complicates the generalization of point-based observations.

To address these limitations, we propose a simplified mass balance approach that captures reach-scale variation in internal CH4 loading without disrupting the system. By comparing CH4 concentrations at two points along a river reach, this method provides an integrated estimate of internal loading while minimizing fieldwork effort. The approach does not require direct access to the river channel, making it suitable for large-scale or long-term monitoring in diverse riverine settings. Despite these advantages, this method has not yet been tested in real river systems.

The objective of this study was to evaluate the feasibility of this mass balance-based method in a river modified by agricultural weir operations. We conducted monthly measurements over a 1-year period at two reaches—upstream and downstream of a weir—within an agricultural river in Japan. We aimed to assess seasonal variation in internal methane loading and to explore how weir-induced changes in hydrology affect methane dynamics. This allowed us to assess the impact of weir construction and operation on internal methane loading. Finally, we compared the results with those obtained from other established approaches to evaluate the reliability and applicability of the proposed method.

Materials and Methods

Study site and sampling

The Kokai River originates from Kokaigaike Pond (elevation: 140 m) in Tochigi Prefecture, flows through Tochigi and Ibaraki Prefectures, and eventually joins the Tone River. The river has a catchment area of 1,043 km2 and a main channel length of 112 km. Land use in the Kokai River basin in 2014 was predominantly mountainous areas (51%), followed by farmland (46%) and residential areas (3%) (National Land Numerical Data, https://nlftp.mlit.go.jp).

Sampling was conducted at four bridges: Nagamine (36.11132°N, 140.00009°E), Shinfukurai (36.07801°N, 140.00464°E), Joso (36.02970°N, 140.02465°E) and Yamato Bridges (N36.01978°N, 140.00563°E) (Fig. 2). We focused on two short reaches, Nagamine Bridge to Shinfukurai Bridge (upstream reach) and Joso Bridge to Yamato Bridge (downstream reach). We applied a mass balance method to determine the internal methane loading within each reach. The flow distances of these reaches were 4.37 and 2.15 km, respectively, and the downstream time (calculated as flow distance divided by mean flow velocities) ranged from 1.63 to 14.0 h for the upstream reach and 0.641 to 2.47 h for the downstream reach. The riverbed areas of these reaches were 0.315 and 0.139 km2. The Fukuoka weir (36.04287°N, 140.02682°E) is upstream of Joso Bridge. The weir is operated for agricultural irrigation during the spring and summer months (April to August), and remains open during the rest of the year. The irrigated area is about 2,800 ha, the maximum water withdrawal is about 13.6 m3 s–1, and the water storage capacity is 2.75 million tons.

Figure 2 Sampling locations (Nagamine, Shinfukurai, Joso, and Yamato Bridges) in the Kokai River, Japan.

The Kokai River is a tributary of the Tone River. Source: GSI Tiles (Seamless Photo) HTML: https://maps.gsi.go.jp.

River water was collected using a stainless-steel beaker with a handle, attached to a rope from the bridges. In the water samples, we measured nutrients (total nitrogen: TN, total phosphorus: TP, total dissolved nitrogen: TDN, total dissolved phosphorus: TDP, nitrate+nitrite: NOx, nitrite: NO2−, ammonium: NH4+, and phosphate: PO43−), dissolved organic carbon (DOC), dissolved methane concentration (CH4), methane oxidation rate and bacterial production (BP). We measured water temperature, conductivity, dissolved oxygen (DO), chlorophyll a concentration, and phycocyanin concentration using a YSI ProDSS water quality probe.

Water samples for dissolved nutrients (TDN, TDP, NOx, NO2−, NH4+, and PO43−) and DOC were immediately filtered through 0.45-µm syringe filters (Durapore, Millipore, Burlington, MA, USA) on the sampling site, and the filtrate was frozen for later analysis.

For measurement of BP, 10 mL of collected river water was poured into 15-mL centrifuge tubes and incubated with 50 nM of [15N5]-2’-deoxyadenosine (15N-dA, NLM-3895-PK, Cambridge Isotope Laboratories, Inc., Tewksbury, MA, USA) in the dark at in situ temperature for 1–5 h depending on water temperature, according to a previous study (Tsuchiya et al., 2015, 2021). The incubation was started at each sampling station immediately after the sampling. The incubation was quenched by adding 99.5% ethanol (Special Grade, Wako, Osaka, Japan) to the sample (final concentration, >20%). After quenching, each water sample was filtered onto a 0.2-µm PTFE membrane filter (Omnipore, Millipore, Burlington, MA, USA), and the filter was stored at –20 °C until further analysis.

For measurement of dissolved methane concentration and methane oxidation rate, we prepared four 20-mL glass vials (GL science) of each station; two out of the four were for T = 0 samples (ambient concentration), and the others were for T = 24 and T = 48 samples in methane oxidation rate measurements. After the sampling, the water samples were immediately transferred to the four glass vials from the stainless-steel beaker. Vials were slowly filled to overflowing with several volumes of water and sealed without any headspace. We added 0.1 mL of 8 mol L–1 potassium hydroxide (KOH) solution (Wako) to two glass vials of T = 0 for sample preservation (Magen et al., 2014). Each glass vial was stoppered with a butyl rubber septum and sealed with an aluminum crimp seal. After sealing, the vials were incubated in the dark. During the field sampling, they were kept in a cooler box filled with ambient river water to maintain in situ temperature conditions. After returning from the field, the incubation was continued in a laboratory incubator set to the in situ river water temperature. After 24 and 48 h, we added 0.1 mL of 8 mol L–1 potassium hydroxide (KOH) solution to the other two vials to quench the methane oxidation.

We used a compact fish sonar to measure river water velocity and depth (Deeper CHIRP +2, Baltic Vision Co., Ltd., Tokyo, Japan). We cast the sonar attached to a fishing rod from the bridge at least three lines in each station and estimated the average flow velocity from GPS information. Depth was measured by moving the sonar along the bridge to determine the cross-sectional area of the river. The discharge was calculated by multiplying the average water velocity and cross-sectional area of the river. Note that the sonar device cannot measure water depths shallower than approximately 15 cm; therefore, extremely shallow shoreline areas were not included in the discharge calculation. However, such areas were spatially limited in our study sites and are expected to have a minimal influence on overall discharge due to their typically low flow velocities (see supporting information, Fig. S1).

Sample analysis

Nutrients were analyzed in our laboratory using a continuous-flow analyzer (QuAAtro, BLTEC) in technical triplicates (Nojiri, 1987; Otsuki et al., 1993). DOC measurements were conducted as nonpurgeable DOC with a TOC analyzer (TOC-V, Shimadzu) equipped with a Pt catalyst on quartz wool. At least three measurements were made for each sample, and analytical precision was typically less than 2%. Potassium hydrogen phthalate (Kanto Chemical) was used as a standard.

To assess bacterial production (BP), bacterial DNA was isolated from the filter samples using the Extrap Soil DNA Kit Plus ver. 2 (J-Bio21, Nippon Steel & Sumikin EcoTech Corp., Chiba, Japan). Cell disruption was achieved through bead beating with a Fast Prep FP120 Cell Disrupter (MP Biomedicals, Santa Ana, CA, USA) at a speed setting of 6.0 for 40 s. DNA was then purified with magnetic bead separation, following the manufacturer’s protocol. Based on prior validation (Tsuchiya et al., 2019), DNA extraction efficiency was assumed to be 100%. Subsequent quantification of 15N-labeled deoxyadenosine (15N-dA) incorporation followed the procedures described in previous studies (Tsuchiya et al., 2015, 2020). Briefly, extracted DNA was enzymatically digested into nucleosides using three enzymes (nuclease P1, Wako; phosphodiesterase I, Worthington Biochemical Corp.; and alkaline phosphatase, Promega Corp., Madison, WI, USA). After the enzymatic hydrolysis, the amount of 15N-dA (15N5-dA + 15N4-dA) incorporated during the incubation was analyzed by LC–MS/MS using 13C1015N5-deoxyadenosine (CNLM-3896-CA, Cambridge Isotope Laboratories, Inc., Tewksbury, MA, USA) as a surrogate (internal standard), with analyses performed in technical duplicate.

The dissolved methane concentration was measured with a gas chromatograph with a flame ionization detector (GC-FID). Before GC injection, we made a headspace of 2 mL helium gas in the sample glass vials and shook them vigorously for 2 min for gas-liquid equilibrium. We injected the 0.5 mL headspace gas to GC-FID. The dissolved methane concentration was then calculated according to a previous study (Magen et al., 2014) using the Bunsen coefficient β for methane at known pressure, temperature, and salinity (Yamamoto, Alcauskas & Crozier, 1976). The relative percent difference (RPD, %) between duplicate dissolved methane measurements at time zero (T = 0) was calculated to assess analytical precision, using the following equation:

(1) R⁢P⁢D⁡(%)=|C1−C2|(C1+C2)/2×100

where C1 and C2 represent the two replicate values. In the present study, the average RPD was 5.2 ± 4.9%, indicating acceptable analytical precision.

Calculation: methane oxidation rate, emission to atmosphere, and mass-balance model

The specific methane oxidation rate was calculated by linear regression of the natural log of methane concentration against time (0, 24, and 48 h). The specific methane oxidation rate is the first-order rate constant for methane oxidation (in units of h–1). The volumetric methane oxidation rate (nmol L–1 h–1) was calculated by multiplying the specific oxidation rate at any station by the measured ambient methane concentration (T = 0). The areal methane oxidation rate (µmol m–2 h–1) was calculated by multiplying the average water depth to the volumetric methane oxidation rate. The precision of methane oxidation rate estimates was evaluated using the standard error (SE) of the regression slope obtained from concentration vs. time plots. The average SE was 30.1 ± 33.1%. A tendency was observed for the SE to increase as the slope decreased, indicating reduced measurement precision at lower oxidation rates.

The percentage of saturation of methane in the water samples was calculated as:

(2) Methanesaturation=CwCeq×100

where Cw is the measured methane concentration in water and Ceq (nmol L–1) is the corresponding equilibrium methane concentration in river water that is in equilibrium with the ambient atmosphere at the in situ pressure and temperature (Wiesenburg & Guinasso, 1979), assuming atmospheric CH4 concentration was 1.9 ppm.

The diffusive flux of methane (Foutgassing, mmol m–2 d–1) from the river surface to the atmosphere was calculated as:

(3) Foutgassing=k×(Cw−Ceq)

where k is the integrated gas transfer coefficient (m s–1) for methane that incorporates physical processes. The Schmidt number (Sc) for CH4 was calculated as a function of in situ water temperature (T, °C), following (Wanninkhof, 1992):

(4) Sc=1,897.8−114.28×T+3.2902×T2−0.039061×T3.

To ensure robustness, five empirical models were used to estimate the gas transfer velocity (k) and its standardized from k600 (i.e., normalized to Sc number of 600 at 20 °C).

The coefficient k was calculated as (Clough et al., 2007):

(5) k=DVh+2.78e−6u102(Sc600)0.5

where DVh is the water current term, which was calculated using the river water velocity (V; m s–1), average river depth (h; m), and a diffusion coefficient for methane in the water (D; m2 s–1) (Jähne, Heinz & Dietrich, 1987). 2.78e−6u102(Sc600)0.5 is a wind term. 2.78e–6 is a conversion factor (cm h–1 to m s–1), α is a constant (0.31), u10 is the wind speed at a height of 10 m above the river, and Sc is the Schmidt number for methane (Wanninkhof, 1992).

The wind speed data (u10) was obtained from nearby weather stations located approximately 9–13 km from the sampling sites, which are available from the Automated Meteorological Data Acquisition System (AMeDAS), operated by the Japan Meteorological Agency (JMA) (https://www.jma.go.jp/jp/amedas/). The k600 were calculated using the following empirical models (Alin et al., 2011; Raymond et al., 2012):

(6) −Ray01:k600=5,037×(V×S)0.89×h0.54

(7) −Ray02:k600=5,937×(1−2.54×Fr2)×(V×S)0.89×h0.58

(8) −Ray05:k600=2,841×V×S+2.02

(9) −Alin03:k600=3.84×10−5+9.72×10−5×V

where S is the average slope (unitless) between bridges derived from DEM data obtained from Geospatial Information Authority of Japan (https://www.gsi.go.jp/). The slopes between Nagamine and Shinfukurai bridges, and Joso and Yamato bridges were 0.000915 and 0.001395, respectively. Fr is the Froude number (Fr = V/(gh)0.5). These models were chosen because they were preferably recommended for streams and small rivers (Raymond et al., 2012). The final gas transfer velocity k (cm h–1) was calculated as:

(10) k=k600×(Sc600)−0.5.

The resulting k values from each model were: Clough: 8.74 ± 3.39, Ray01: 23.0 ± 21.8, Ray02: 26.3 ± 24.1, Ray05: 12.7 ± 4.6, Alin03: 24.6 ± 9.5 (mean ± SD; units: cm h–1). The diffusive methane flux was calculated separately using each model. The standard deviation of the five flux estimates was then used as an indicator of model-based uncertainty. This approach provides a quantitative assessment of variability among widely used gas transfer models, as recommended for small river systems.

The obtained methane concentrations, oxidation rates and outgassing fluxes were combined into a simple mass-balance model to estimate the internal methane loading (Fig. 1). Here, concentration changes of methane between upstream and downstream (Nagamine~Shinfukurai bridges and Joso~Yamato bridges) were modeled based on: (a) the external loading of methane via such as tributary and groundwater inflow (Fexternal; mol m–3 d–1), (b) the internal methane loading mainly produced in river bed (Finternal; mol m–2 d–1), (c) methane oxidation in water column (Foxidation; mol m–3 d–1), and (d) methane emission to atmosphere (Foutgassing; mol m–2 d–1) as:

(11) Cdownstream−Cupstream=∫t1t2⁡Fexternaldt+AV∫t1t2⁡Finternaldt−∫t1t2⁡Foxidationdt−AV∫t1t2⁡Foutgassingdt

where C is dissolved methane concentration (nmol L–1), t is streaming time (h) calculated by streaming distance (I; m) divided by velocity (u; m s–1), A is riverbed area (m2) and V is water volume between the bridges (upstream and downstream) (Fig. 1). The Finternal refers to the net rate of methane supply to the river water, including methane production and oxidation processes in the riverbed. If the reach is sufficiently short such that the discharge at the upstream and downstream ends are nearly identical, Fexternal can be assumed to be zero. This assumption allows Finternal to be estimated using a mass balance approach. Since the flow velocity at Shinfukurai Bridge on August 23, 2023, could not be measured due to the wind at the time of the observation, the calculation assumed that the flow velocity at Shinfukurai Bridge was the same as that of the upstream Nagamine Bridge. In the present study, Finternal was calculated assuming that CH4 was uniformly loaded from the riverbed within the section. The uncertainty of the internal CH4 loading estimate (Finternal) was calculated by propagating the independent errors associated with CH4 oxidation rates (Foxidation) and diffusive fluxes to the atmosphere (Foutgassing). Specifically, the standard error of Finternal was derived using the root-sum-square method, assuming the errors in Foxidation and Foutgassing were uncorrelated. To ensure the validity of Finternal estimation, data were excluded when the relative difference in discharge between upstream and downstream sites exceeded 30%, as this may indicate substantial external inflows or exceed expected measurement uncertainty. For all other cases, we assumed that the discharge remained approximately constant within the reach, considering typical measurement uncertainty.

Statistical analysis

Partial least squares (PLS) regression was used to identify significant drivers (independent X variables) of internal methane loading (Finternal, dependent Y variable). PLS was chosen because of its robustness against multicollinearity among X variables and its insensitivity to deviations from normality (Wold, Sjöström & Eriksson, 2001). The X variables included in the analysis were water velocity, average depth, discharge, water temperature, DO, PO43−, NH4+, NO3−, DOC, chlorophyll a, and BP. Before running the PLS regression, data were mean-centered and scaled to unit variance to mitigate the effects of differing measurement units and ensure numerical stability. To assess the importance of the independent X variables in the model, the variable influence on projection (VIP) score was used. Independent X variables with VIP > 1.2 were considered significant. PLS regression was conducted using JMP 14.3.0 (SAS Institute Inc., Cary, NC, USA).

Results

Environmental variables

The average water depth in the upstream reach (Nagamine and Shinfukurai bridges) ranged from 2.8 to 3.5 m between April and August, exceeding the depth in the downstream reach (1.1 to 2.1 m) during the same period (Fig. 3A). This difference is attributed to the operation of the Fukuoka weir for irrigation, which increased water depth upstream. Similarly, flow velocity was slower in the upstream reach during the irrigation period (April to August, Fig. 3B). On September 26, 2022, both water depth and flow velocity were unusually high due to flooding. The discharge for each month was almost the same between the two bridges in each upstream and downstream reach (Fig. 3C). In certain months (June 2022 in upper reach and September 2022 and May 2023 in lower reach), the discharge difference between upstream and downstream sites exceeded the 30% threshold. For these months and reaches, Finternal values were not calculated. In all other months, discharge differences were within the assumed uncertainty range, and Finternal was estimated accordingly.

Figure 3 Seasonal variations of (A) average water depth, (B) average water velocity and (C) discharge at four bridges in the Kokai River.

No data is available for the Joso Bridge in July 2022, Nagamine and Shinfukurai bridges in August 2022, and Shinfukurai Bridge in August 2023. The yellow background in the figure indicates the irrigation period (April to August).

Water temperature varied from 1.1 °C (January) to 29.1 °C (August; Fig. 4A). Dissolved oxygen (DO) concentration was lower in the upstream reach during the irrigation period, while NH4+ and PO43– concentrations were higher upstream during the same period (Figs. 4B and S2A, S2B). Specific conductivity was higher during the non-irrigation period (maximum: 231 µS cm−1) and lower during irrigation (minimum: 163 µS cm–1) (Fig. S2A). Seasonal trends in nitrate (NO3−) concentrations showed higher values in winter and lower values in summer (Fig. 4C). Dissolved organic carbon (DOC) concentrations ranged from 1.2 to 2.3 mgC L–1 (Fig. S3A). Bacterial production (BP) was lowest in winter and increased in summer, with no clear differences between sites (Fig. S3B). Chlorophyll-a (Chl) concentrations were generally higher in the downstream reach during irrigation (Fig. S3C). The complete dataset, including all raw measurements of environmental parameters and methane-related variables, is provided in File S1.

Figure 4 Seasonal variations in (A) water temperature, (B) dissolved oxygen concentration, and (C) nitrate (NO3− -N) at four bridges in the Kokai River.

The yellow background in the figure indicates the irrigation period (April to August).

Methane dynamics

Dissolved methane concentrations fluctuated between 237 ± 19 and 1,271 ± 6 nM (Fig. 5A). Within a given month, methane concentrations tended to decrease downstream. However, higher concentrations were sometimes observed during the irrigation period at Shin-fukurai Bridge compared to Nagamine Bridge in the upper reach. In the lower reach, methane concentrations at Joso Bridge were consistently higher than at Yamato Bridge in both irrigation and non-irrigation periods. The specific methane oxidation rate ranged from –0.00165 ± 0.00046 to 0.0328 ± 0.00105 h–1, with relatively high values observed in downstream reaches such as Joso and Yamato during the summer irrigation period. Areal and volumetric methane oxidation rates peaked at 66.4 ± 18.5 µmol m–2 h–1 and 23.2 ± 6.5 nmol L–1 h–1, respectively (Fig. 5B). Notably, during the irrigation period, areal methane oxidation rates were higher in the upstream section (Nagamine and Shin-fukurai: 16.2 ± 18.1 µmol m–2 h–1) than in the downstream section (Joso and Yamato: 5.97 ± 4.68 µmol m–2 h–1), likely due to greater water depth in the upstream reaches. In contrast, from October to march, the areal oxidation rates showed little spatial variation, with an overall average of 1.1 ± 1.3 µmol m–2 h–1. Methane diffusive emission fluxes to the atmosphere ranged from 199 ± 149 to 31.6 ± 9.6 µmol m–2 h–1 (Fig. 5C). During the irrigation season, diffusive fluxes increased in both upstream (Nagamine and Shin-fukurai: 88.9 ± 26.7µmol m–2 h–1) and downstream reaches (Joso and Yamato: 107 ± 44 µmol m–2 h–1). From October to March, the values remained low throughout the river (mean: 53.0 ± 12.7 µmol m–2 h–1).

Figure 5 Seasonal variations in (A) methane concentration, (B) methane oxidation rate, and (C) methane diffusive flux to atmosphere at four bridges in the Kokai River.

The yellow background in the figure indicates the irrigation period (April to August).

Internal methane loading (Finternal) varied from –33.1 ± 20.7 to 160 ± 46 µmol m–2 h–1, with a mean of 61.3 ± 50.8 µmol m–2 h–1 (Fig. 6). In both the upper and lower reaches, Finternal tended to increase during the summer and decrease during the winter, indicating a consistent seasonal pattern. No significant difference in Finternal was found between the upper reach (66.0 ± 51.6 µmol m–2 h–1) and the lower reach (56.2 ± 51.4 µmol m–2 h–1) throughout the year (Welch’s t-test, n = 27, p > 0.05). However, during the latter half of the irrigation period (July to August), Finternal was significantly higher in the upper reach (149 ± 11 µmol m–2 h–1)than in the lower reach (92.4 ± 23.8 µmol m–2 h–1) (Welch’s t-test, n = 7, p = 0.0112).

Figure 6 Seasonal variation in methane flux from sediment to water (Finternal) in the upper and lower reaches of the Kokai River.

Seasonal variations of internal methane flux (Finternal) from sediment to water in the reaches between Nagamine and Shinfukurai Bridges (upper reach) and between Joso and Yamato Bridges (lower reach) in the Kokai River.

In the PLS regression analysis, 58.2% of the variation in Finternal was explained by the explanatory variables (R2 = 0.582, n = 27, p < 0.001). Average water depth, water temperature, DO and NO3– were selected as explanatory variables with VIP > 1.2 for the dependent variable of Finternal (Table 1). The estimated coefficients of DO and NO3– were negative, whereas those of average water depth and water temperature were positive.

Table 1 Model coefficients for centered and scaled data in partial least square (PLS) regression for internal methane loading (Finternal) of the Kokai River.

Predictors	Coefficient	VIP	
Velocity	0.021	0.20	
Depth	0.126	1.20	
Discharge	0.107	1.02	
Temperature	0.147	1.39	
DO	−0.132	1.26	
PO4	−0.085	0.81	
NH4	−0.090	0.85	
NO3	−0.152	1.44	
DOC	0.047	0.44	
Chl a	0.045	0.43	
BP	0.112	1.07	
Intercept	0		
Note:

Predictors with variable importance in projection (VIP) greater than 1.2 are shown in bold.

Discussion

In this study, we estimated the internal methane loading using a mass balance approach based on the difference in methane concentration between two locations (upstream and downstream) in a river for the first time. The internal methane loads in the upstream section of the weir showed higher values than the downstream section. Correspondingly, during the weir operation period, DO concentrations were lower, while NH4+ concentrations were higher in the upstream section, where flow velocity was low and water depth was high (Figs. 3, 4 and S2). This method effectively captured internal methane loading variations, reflecting differences in hydrological and water quality environments.

The internal methane loading (Finternal) estimated in this study ranged from –33 to 160 µmol m–2 h–1, with an average of 61 ± 51 µmol m–2 h–1 across the two reaches (Table 2). These values fall within the range reported in recent river studies that used comparable mass balance, stable isotope and modeling approaches (Rovelli et al., 2022; Chen et al., 2023; Michaelis et al., 2024; Balathandayuthabani et al., 2024), which generally reported values between –0.25 and 1,313 µmol m–2 h–1. In contrast, fluxes measured using benthic chambers tend to be higher (up to 2,427 µmol m–2 h–1), likely reflecting localized hotspots or short-term maxima (Rovelli et al., 2022). While the dynamics of methane production differ between flowing and lentic systems, the range of Finternal values observed here is also comparable to that reported for the littoral zones of shallow lakes (Rudd & Hamilton, 1978; Nakamura et al., 1999; Bastviken et al., 2008), where benthic methanogenesis under anoxic conditions similarly contributes to CH4 fluxes. Although caution is needed when comparing across system types, this consistency suggests that the magnitude of sedimentary CH4 input observed here is reasonable.

Table 2 Previously reported CH4 internal loading (CH4 net flux from sediment to water) in lakes and rivers.

Type	Location	Methods	CH4 internal loading	Reference	
(µmol m−2 h−1)	
Lake	Lake 227, Canada	Mass balance	33.3	Rudd & Hamilton (1978)	
	Lake Kasumigaura, Japan	Mass balance	99.9	Nakamura et al. (1999)	
	Lakes Paul, Peter and Hummingbird, Wisconsin	Mass balance	100~133	Bastviken et al. (2008)	
River	Avon River, UK	Benthic chamber	24~2,427	Rovelli et al. (2022)	
		Mass balance	–0.25~59		
	Columbia River, Washington	Peeper and model	0~100 (outlier 833)*	Chen et al. (2023)	
	Moosach River, Germany	Peeper and model	1.3~4.6	Michaelis et al. (2024)	
	Skogaryd Research Catchment (SRC), Sweden	Peeper, stable isotope and mass balance	0.033~1,313	Balathandayuthabani et al. (2024)	
	Forsmark catchment, Sweden	Peeper, stable isotope and mass balance	3.3~354		
	Kokai River, Japan	Mass balance
(difference within reach)	–33~160	This study	
Note:

* Negative values (methane production rate is less than methane oxidation rate) were reported.

To evaluate the plausibility of our estimated internal methane loading (Finternal), it is useful to compare our values with methane production rates derived from sediment incubation studies. For instance, Romeijn et al. (2019) reported methane production rates from incubated river sediments of different compositions: 10.5 ± 15.6 µmol CH4 m–2 h–1 for chalk sediment and 1.33 ± 2.28 µmol CH4 m–2 h–1 for sandstone. These values are within the same order of magnitude as the Finternal rates reported in our study. It should be noted, however, that such incubation experiments often use surface sediments with relatively higher redox potential compared to deeper, more anoxic layers that are likely responsible for in situ methane generation. Therefore, actual subsurface production rates may be higher under natural hypoxic conditions. Nonetheless, the general agreement in magnitude between literature values and our flux-based estimates supports the credibility of our method for estimating internal CH4 inputs.

The presence of the Fukuoka weir and its associated water withdrawal significantly influence the hydrological and biogeochemical conditions of the upstream reach. During the irrigation period (April to August), the weir operation leads to increased water depth and reduced flow velocity upstream, effectively enhancing hydraulic residence time. These conditions promote sediment deposition and the development of anoxic zones near the sediment–water interface, thereby facilitating methane production within the benthic environment (Bednařík et al., 2017). This hydrological alteration is consistent with the seasonal trend observed in our data, where Finternal values were markedly higher upstream of the weir during the latter half of the irrigation period. The prolonged water residence time likely allows for greater accumulation and microbial processing of organic matter, which in turn fuels methane generation. These observations highlight the role of hydraulic structures such as weirs in modulating in-stream greenhouse gas dynamics, especially in agricultural river systems where irrigation infrastructure is common. Future studies could further investigate the longitudinal impact of such structures by comparing reaches with and without flow regulation under similar land use and geomorphological conditions.

In the present study, the internal methane loading sometimes showed negative values, which was the same as in the two previous studies on rivers (Rovelli et al., 2022; Chen et al., 2023). As the internal methane loading is an apparent value, the negative values suggest that the methane oxidation rate in the riverbed is greater than the supply of methane from the riverbed. Although the present study measured the methane oxidation rate in river water but not in the riverbed, it is known that methane oxidation in the riverbed is relatively high compared to that of river water (Shelley et al., 2015). The biogeochemical model in the Columbia River showed that the methane oxidation rate exceeded the methane production rate in riverbed sediment when the vertical hydrologic exchange flows (either upwelling or downwelling) were high (Chen et al., 2023). Since negative methane flux from sediment has not been reported in lakes (Bastviken et al., 2008), methane oxidation in the riverbed may be a distinctive feature of rivers that helps mitigate methane emissions. In addition, the negative values of internal methane loading suggest that methane oxidation in the riverbed is highly effective. Identifying such river sections and analyzing their hydrological and hydraulic characteristics could provide valuable insights for applying river engineering techniques to reduce methane emissions.

The ratio of methane oxidation to diffusive emission (Foxidation/Foutgassing) provides insight into the relative importance of microbial CH4 consumption before it escapes to the atmosphere (Fig. 7). Throughout most of the year, the ratio remained below 0.1 in both upstream (Nagamine–Shin-fukurai) and downstream (Joso–Yamato) reaches, indicating that only a small fraction of dissolved CH4 was oxidized before emission. Notably, the upstream reach consistently showed higher Foxidation/Foutgassing ratios than the downstream section during the irrigation season. This spatial pattern may reflect differences in hydraulic retention and characteristics that favor CH4 oxidation in the upstream area. These findings imply that while CH4 oxidation is generally a minor process compared to emission in this system, its temporal and spatial variability can substantially influence net CH4 flux and should be considered when evaluating the effectiveness of CH4 sinks in fluvial environments. Seasonal variation in internal CH4 loading (Finternal) was clearly observed in this study, with significantly higher values during summer months. This variation can be attributed to several interacting environmental factors, as supported by our PLS regression analysis. Notably, water temperature and depth were positively associated with Finternal, while dissolved oxygen (DO) and nitrate (NO3–) concentrations showed negative associations. Water temperature is a key driver of microbial methane production. Methanogenesis is known to be highly temperature-dependent, typically exhibiting a Q10 value of approximately 4.1, which is higher than that of methane oxidation (Q10 ≈ 2) (Segers, 1998). This difference implies that a 10 °C increase in temperature can roughly quadruple methane production, whereas oxidation rates would only double. Thus, elevated summer temperatures likely stimulated microbial activity in the sediments, enhancing CH4 production and resulting in higher Finternal. The observed negative correlation between DO and Finternal is consistent with the known inhibitory effect of oxygen on methanogenesis, as CH4 production occurs under strictly anoxic conditions (Segers, 1998; Conrad, 2007). Similarly, the inverse correlation with NO3– may reflect its role as an alternative electron acceptor that can suppress methane production by outcompeting methanogens for electron donors such as acetate and H2 (Achtnich, Bak & Conrad, 1995; Stanley et al., 2016). Depth was positively associated with Finternal, reflecting structural differences between upstream and downstream reaches. In the upstream section, the presence of the weir increases water depth during the irrigation period, leading to reduced flow velocity and prolonged residence time, which further enhances conditions favorable for methane accumulation and release. In general, low flow velocity and high water depth make it easier for sediment to accumulate, which results in favorable conditions for methane production (Bednařík et al., 2017). Together, these findings provide a mechanistic explanation for the seasonal trends observed in Finternal and underscore the role of temperature and oxygen availability as primary controls on sedimentary CH4 dynamics in fluvial systems.

Figure 7 Seasonal variation in the Foxidation/Foutgassing ratio in the upper and lower reaches of the Kokai River, measured between April 2022 and September 2023.

Seasonal variation in the ratio of methane oxidation flux (Foxidation) to atmospheric diffusive flux (Foutgassing) in the reaches between Nagamine and Shinfukurai Bridges (upper reach) and between Joso and Yamato Bridges (lower reach) in the Kokai River. Values greater than 1 indicate net removal of CH4 via oxidation exceeding atmospheric emission, while values less than 1 suggest dominant emission.

The estimation of internal CH4 loading (Finternal) in this study is based on a mass balance approach that assumes minimal external CH4 input from sources such as tributaries or groundwater (i.e., Fexternal ≈ 0). This assumption is supported by several observations. First, the selected river reaches do not include any significant tributary inflows, minimizing the likelihood of external surface CH4 inputs. Second, discharges measured at upstream and downstream points showed negligible differences throughout the year, suggesting the absence of substantial lateral water input. Third, regional hydrogeological data indicate that the study area is characterized by shallow alluvial deposits and homogeneously flat agricultural land use, which are typically associated with weak hyporheic exchange and limited vertical groundwater upwelling (Magliozzi et al., 2018). While direct measurements of groundwater CH4 flux were not conducted, the lack of evidence for significant groundwater–surface water interaction in this geomorphologically simple system supports the validity of the assumption. Nevertheless, we acknowledge that unmeasured localized groundwater discharge may introduce small CH4 inputs under certain hydrological conditions. Future studies incorporating porewater profiles, tracer techniques, or direct seepage measurements would help constrain Fexternal more robustly.

The spatial resolution required for the application of this method depends largely on the analytical precision of dissolved CH4 measurements. In our study, the relative percent difference (RPD) for duplicate methane concentration measurements averaged 5.2%, with a maximum of 22.8%. For example, assuming a typical CH₄ concentration of 350 nM in the lower reach, a detectable concentration change should exceed 18.1 nM (mean RPD) or 79.8 nM (maximum RPD) to ensure reliability in estimating internal CH4 inputs. Considering only diffusive loss to the atmosphere (excluding oxidation due to its strong seasonal variability), we estimated that with an average CH4 emission flux of 66 nmol L–1 h–1, the river water would need to travel at least 0.27 h (for 18.1 nM loss) or up to 1.2 h (for 79.8 nM loss) to generate a detectable concentration decrease. With an average flow velocity of 0.41 m s–1, this corresponds to a flow path length of approximately 405 m (minimum) to 1,771 m (maximum). The actual distance between measuring sites in the lower reach (Joso–Yamato) was 2.15 km, which exceeds the minimum requirement based on analytical precision, thus validating the suitability of our setup. In contrast, the maximum allowable distance is constrained by hydrological consistency. As long as no significant inflow, outflow, or groundwater exchange alters the discharge between sites, longer distances may be acceptable and even desirable to integrate fluxes over broader scales.

To directly address the feasibility of the mass balance-based approach introduced in this study, we evaluated its performance, applicability, and limitations. This method estimates internal CH4 loading (Finternal) by balancing CH4 oxidation rates with diffusive emissions, providing a practical and efficient framework particularly suited to river systems where sediment-derived methane is a dominant source and hydrological conditions are relatively stable. The method was applied across multiple seasons and spatial reaches, yielding Finternal values in line with sediment methane production rates reported in the literature (e.g., Romeijn et al., 2019). However, several limitations must be acknowledged. First, the method assumes quasi-steady-state conditions among CH4 oxidation, emission, and internal input, which may be violated during storm events or rapid hydrological shifts. Second, uncertainty in Finternal arises from compounded variability in CH4 oxidation rates and diffusive flux estimates. We addressed these uncertainties by propagating errors from both measurements and by using five different empirical models to calculate the gas transfer velocity (k) (Clough et al., 2007; Alin et al., 2011; Raymond et al., 2012), thus accounting for model-related variability. Despite these limitations, the method is advantageous due to its moderate field requirements and ease of application, especially in systems where direct measurements of sediment CH4 production (e.g., via incubations or isotope-based approaches) are not feasible. Further validation using complementary techniques, such as benthic chambers or isotope mass balances, could improve robustness. Nevertheless, the approach offers a promising tool for exploring spatial and seasonal variation in internal CH4 sources in modified riverine systems.

Conclusions

We estimated the methane flux from riverbed sediment to water (internal methane loading) using a mass balance approach based on the difference of methane concentrations between upstream and downstream where the discharge change was negligible. Observations of upstream and downstream areas of the weir revealed that internal methane loading was higher in the upstream area, which provides favorable conditions for higher methane loading from the riverbed. The result demonstrates that the proposed method can capture changes in internal loading in response to environmental variations. Although this method is limited to sections with minimal discharge variations, its simplicity enables measurements across multiple sections. By integrating estimates from river reaches, the proposed method is applicable to large-scale assessments of internal methane loading in river systems, allowing for the identification of methane flux hotspots (both of higher and lower methane loading) from riverbeds and the assessment of total internal methane loading in river systems. By examining the geographical, hydraulic, and hydrological characteristics of reaches where Finternal is low or negative, we can obtain insights that contribute to future river management strategies. If internal methane loading can be modeled in the future, it will become possible to estimate external methane loading, such as groundwater inputs, in river sections with discharge variations. This would improve our understanding of methane dynamics in river systems.

Supplemental Information

Supplemental Information 1 Raw data used for methane flux estimation and related parameters in the Kokai River.

Supplemental Information 2 River cross-sectional profile at Nagamine, Shin-fukurai, Joso and Yamato Bridges in July 2023.

Supplemental Information 3 Seasonal variations in (a) specific conductivity, (b) ammonium concentration (NH4+-N), and (c) phosphate (PO43+-P) at four bridges in the Kokai River.

Supplemental Information 4 Seasonal variations in (a) dissolved organic carbon, (b) bacterial production, and (c) chlorophyll a concentration at four bridges in the Kokai River.

The authors declare that they have no competing interests. The authors used ChatGPT (OpenAI) to improve the clarity and grammar of the English in the manuscript. All content was reviewed and edited by the authors.

Additional Information and Declarations

Competing Interests

The authors declare that they have no competing interests.

Author Contributions

Kenji Tsuchiya conceived and designed the experiments, performed the experiments, analyzed the data, prepared figures and/or tables, authored or reviewed drafts of the article, and approved the final draft.

Shingo Miura performed the experiments, authored or reviewed drafts of the article, and approved the final draft.

Ayato Kohzu performed the experiments, analyzed the data, authored or reviewed drafts of the article, and approved the final draft.

Data Availability

The following information was supplied regarding data availability:

The raw data are available in the Supplemental File.

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
