# Peer review of "Estimating internal dissolved methane loading in rivers using a mass balance approach"

_PeerJ, doi:10.7717/peerj.20238_

## Round 0.1 · original submission · Major Revisions

· Academic Editor

Major Revisions

Two reviews of your manuscript are enclosed. You are invited to address the issues raised by the reviewers in a revised manuscript. Please respond to each comment, point by point, referencing page or line numbers in the revised manuscript file.

Both reviewers recommend moving content from the supplemental file to the main text. They also highlight a need for additional details about the precision, uncertainty and variability of data. Reviewer 1 makes that request in the context of more fully evaluating limitations and applicability of the method. Similarly, Reviewer 2 requests analysis of uncertainty and errors associated with the methane loading values in order to evaluate whether differences between reaches can be assessed as significant.

Reviewer 1 provides both written comments and an annotated copy of the manuscript. The reviewer raises questions about minimum and maximum distances between measurement points along the rivers, details about the ecological context of methane cycling in rivers, the basis for assumption that F_ext is negligible, information about groundwater input to the reaches studied, and the potential impact of the weir on the study results.

Reviewer 2 indicates a need to improve referencing in the manuscript. The reviewer also requests addition of a figure showing sites and sampling locations, and a clearer explanation of study design and methods.

Reviewer 1 ·

Basic reporting

The language of the manuscript is clear and well written.
Introduction and Literature references are sufficient; and with the relevant background.
The figures are well preapared. However, I would recommend to include the figure of the parameters: discharge, water temperature and NO3 into the main text, not as supplement.
Data table is ok

Experimental design

The hypotheses are well defined.
The method applied seems to be promising, however its applicability and limitations are not sufficiently discussed. Also, the findings of the study (methane flux) should be discussed in a more ecological context.
Some details of the methods could be described in more detail (see comments in the ms file). Also, no information on precision or variability of the data is shown.

Validity of the findings

A Discussion of the method has to be added. How can you assure that the F_ext is negligible? Is there any information of groundwater input to this river? If you would incubate the river sediment, the production rate obtained should be than equivalent to your F_int, are there any data available, also from literature?
The authors should also discuss the further application of the method, what would be minimal or maximal distance between two measuring points? What are the limiatations of the method?

The influence of the weir and its water withdrawal should be discussed in more detail.

Please add also a discussion on the diffusive fluxes, why are they higher for the Nagamine site? While the other stations range between 0.5 - 1, thus rather similar….?

The hypotheses from the introduction were:
….To evaluate the feasibility of this mass balance-based method in a river modified.
There is no clear discussion of the method and ist feasybilligy as outlined above.

…..We aimed to assess seasonal variation in internal methane loading and to explore how weir-induced changes in hydrology affect methane dynamics.
The manuscript describes the seasonal variation, yes, but the discussion oft he reasons for these variations is meagre.

……This allowed us to assess the impact of weir construction and operation on internal methane…
I cannot find a clear assessment of the influence of the weir.

Annotated reviews are not available for download in order to protect the identity of reviewers who chose to remain anonymous.

Reviewer 2 ·

Basic reporting

The language is Ok. The background of the story is sufficiently developed. Referencing could be improved. There are studies using isotopes to infer CH4 sources in streams. Balathandayuthabani et al (https://agupubs.onlinelibrary.wiley.com/doi/pdf/10.1029/2023JG007836) for example combined different methods including mass balances (see also references in that publication).
The figures are OK. The authors placed a large part of their data in the supplement. I wonder, since this manuscript is not very long, if some of the figures in the supplement could be moved to the main text. At least having a map with the positioning of the reaches relative to the dam would help a lot to understand the study design. I had problems to figure out what was where.
l. 99. They mention “downstream time”. Is that the residence time of the water in the respective reach? How was that quantified?
l.115: explain abbreviation BP
l.l286: If Rovelli et al did a mass balance this contradicts l.60-68.

Experimental design

I recommend to bring all the process rates to the same unit (e.g. total rates for each reach, unit would be mol/d or similar). This would enable comparing the relative importance of different processes (compare total loading with total oxidation and outgassing). They even use different units for similar things (like e.g. In figure 3 and figure 2c or in l.260 and l.261), which make it difficult to judge the relative importance of the different processes.
I am missing quantification of uncertainty. The figures, for example, do not contain any information about uncertainty or variability. This is especially relevant in figure 3, which is calculated as a difference between large numbers each coming with uncertainty.
They measured stream hydraulics using GPS information. How was that done? Using drifters? I wonder if there are no available official gauging data for comparison? Also: How are shallow stream areas at the shore considered in the discharge quantification? It would also be good to report stream width (to better judge how important wind speed was for k600).
l.174: I guess you calculated the rate by multiplying a first order rate constant with concentration. Please differentiate between first order rate constant and rate in the text.
l.187: Is Clough et al 2007 the correct original reference for the equation? Why was this parametrization chosen?
l.194: How far are the weather stations from the study reaches.

Validity of the findings

They show that the rather easy and handy method gives reasonable numbers. However, before really interpreting the calculated numbers as methane loading, some kind of uncertainty and error analysis is necessary. Otherwise it is not possible to judge e.g. differences between the reaches.
l.278-280: How is this statement supported by data? You show internal loading variations – that is true – but to judge whether the method is OK you would need a reference method or at least an error and plausibility analysis.
CH4 concentrations in streams are controlled locally (see e.g. Stanley et al 2016). In the methods section they report landuse in the catchment to be 46% farmland. From the map in the supplement it looks as if both reaches are in a 100% farmland area.
l.281-292: This section is rather long and not well structured. I wonder if it makes much sense to compare with data measured in lakes.

Additional comments

This is a nice and focused case study. However, before it can be accepted it needs to be improved with respect to error quantification and data presentation.

---

## Round 0.2 · Minor Revisions

· Academic Editor

Minor Revisions

The authors have made extensive revisions made to the manuscript, which satisfactorily address issues raised by reviewers. Specifically, information has been moved to Supplemental Content to improve manuscript clarity, a detailed analysis has been conducted of data uncertainty, methodological assumptions have been clarified, the discussion has been substantively expanded to examine methane data in the context of environmental and ecological parameters, and referencing has been improved.

The manuscript is acceptable for publication pending several minor changes to the introduction section that are outlined below.

Line 53 - ‘understood’ = ‘quantified’

Lines 55-56 - “Reducing methane emissions from rivers to the atmosphere is crucial for mitigating climate change.”

>> This sentence is not justified by the preceding statements. If methane emissions from rivers account for <5% of total global emissions to the atmosphere, it is unclear why reducing this flux source is crucial for mitigating emissions relative to large sources, in particular, anthropogenic sources. It is more accurate to state that reducing methane emissions from rivers ‘has potential to contribute to mitigating climate change’.

Line 174 - is the word ‘mobbing’ meant to be ‘moving’

---

## Round 0.3 · accepted · Accept

· Academic Editor

Accept

The manuscript is accepted in its current form.